# The Relationship between Narrative Skills and Executive Functions across Childhood: A Systematic Review and Meta-Analysis

**DOI:** 10.3390/children10081391

**Published:** 2023-08-15

**Authors:** Nicoletta Scionti, Laura Zampini, Gian Marco Marzocchi

**Affiliations:** Department of Psychology, University of Milan-Bicocca, 20126 Milan, Italylaura.zampini1@unimib.it (L.Z.)

**Keywords:** executive function, narrative, language, working memory, shifting, planning

## Abstract

Executive functions (EFs) and narrative competence (NC) are two important predictors of many outcomes in human development. To date, however, it is unclear whether these skills develop synergistically—supporting or opposing each other—or whether they are independent of each other. The purpose of this meta-analysis is to understand if these skills are related to over development and if the magnitude of their association changes over time; differs in typical and atypical development; and changes with EF (inhibition, working memory, flexibility, planning) and NC (oral, written; micro and macrostructural level). For this purpose, 30 studies containing 285 effect sizes were selected and combined. The results show that EFs and NC are weakly associated with each other (r = 0.236, *p* < 0.001) and that this association decreases with age (*b*(267) = −0.0144, *p* = 0.001). They are more associated in preschool and early elementary school grades, becoming more independent after seven years old. Between 3 and 7 years of age, the association seems stronger in atypically developing children and for macrostructural NC. Additionally, before 7 years old, the various EF domains seem to associate indistinctly with NC, and only later specific links between EFs and NC would be observed.

## 1. Introduction

Narratives represent an essential device for human communication and are a vehicle for cultural transmission.

The onset of the use of narratives represents a critical step in studies of language development, where storytelling represents a real and contextualized request for children. Therefore, it is seen by many authors as a “naturalistic” approach to studying language development [1]. Evaluation of children’s oral narratives is of significant interest to researchers and practitioners, as being a proficient narrator is an important skill in the life of children, and in adults. Oral narrative skills are a key component of most school curricula, and several studies support the importance of narrative abilities to academic and social success for both typically developing children and children with language and learning disabilities [2,3]. Extant research reports that good narrative skills are positively associated with structural language, literacy, and social skills [4,5,6].

Telling stories is a multi-componential complex competence. It requires the child to be able to plan and execute their production of the story’s plotline by using appropriate vocabulary, grammar, and syntax. Studies on the development of narrative skills have identified that stories have a typical structure, or story grammar [7], following a “schema” that children and adults use to understand, interpret, and produce stories. According to Stein and Glenn’s [7] story-grammar model, stories must include a setting and an episode system at a minimum. An episode consists of an introduction, a provision of the setting and description of the characters in the story, a problematic situation that shapes the protagonist’s goal, attempts to solve the problem, and a conclusion (e.g., [7,8,9]). Stories may also include multiple episodes organized in a linear or a hierarchical manner, resulting in more complexity (e.g., stories with multiple embedded episodes within a particular story arc). Developmental studies reveal that the acquisition of narrative proficiency is a slow process, which emerges in the preschool years and is not fully developed until adulthood [10]. In early childhood, there is a disproportionate emphasis on characters’ actions in narratives without a link to the plot line [11]. At 2 years, narratives are descriptions of character actions, and labels posited without a link to a central theme. Between 3 and 4 years, narratives generally include some local connections between adjacent story events and simple inferences across the story episode. At 4 years, children begin to use structural components of narratives, which generally include causal connections between events. However, until 5, children still show difficulty conceiving an overall plot or overarching goal. 

It is not until 6–7 years old that children are able to produce “true narratives”. At this age, their narratives follow a logical progression of events, including sub-plots and understanding of time frames. After 7 years old, narratives are generally well-structured. Progress in literacy acquisition seems to play a significant role in this passage. The narrative generation process is thought to draw critically on reading skills. For example, Abbott and Berninger [12] found that reading contributes significantly to the quality of narrative composition for children in the first three grades.

Empirical findings suggest that reading and writing draw on shared knowledge yet are separate skills with distinct developmental trajectories [13,14]. In a study with 120 third-grade children, Olinghouse [15] found that reading skills directly influenced compositional quality. There are aspects of continuity and discontinuity in the transition from oral to written narrative composition during this period. Studies on typically developing children provide evidence that children who master writing preserve their narrative skills in the transition between the codes [16]. However, for those children who do not master it efficiently (e.g., children with learning disabilities and other neurodevelopmental disorders), written narrative composition becomes an obstacle.

At 8–10 years, children generally manage structural components correctly and demonstrate that they know how to tell a story to another person. After 10 years, narratives are more complex, detailed, and structurally coherent. Children use various linking devices (e.g., prepositions, conjunctions, and adverbs) and demonstrate more effort to engage the listener’s attention and adapt to different audiences.

Across development, oral and written narratives can be studied at the macro- and microstructure levels. Microstructure refers to specific features of the language used to convey ideas, including the use of decontextualized language and grammatical complexity (e.g., [17,18]). In contrast, macrostructure refers to global narrative features, particularly the ability to produce a story that is overall well structured, coherent, and cohesive. During development, a remarkable increase involves the macrostructural level (e.g., [19]), particularly in the transition from preschool- to school-age (e.g., [20,21]). This period is characterized by the rapid qualitative increase in executive functions (EFs).

EF refers to a broad set of neurocognitive processes underlying goal-directed control of thought, behaviour, and emotion that allow for adaptation to environmental demands [22]. Like narrative skills, EFs are predictors of great relevance to many developmental outcomes. A large body of research has demonstrated substantial links between EFs and academic achievement, literacy, health, wealth, and criminality [23] in children of various ages with and without neurodevelopmental disorders (see [24,25] for reviews).

There is no unanimous agreement on which domains include the construct of EFs.

Scholars studying EFs deal with the problem that EFs are initially unitary or undistinguishable (e.g., [26]), but they differentiate across development. To date, when and how they differentiate is still unclear. In the adult population, three specific core domains were identified: inhibition, updating of working memory, and shifting [27]. This finding was replicated in research with 8- to 13-years-old children [28]. However, research with younger children usually yields a smaller number of factors. Especially for preschool age, the debate on the structure of EFs is still open. This period is the most critical for the rapid changes occurring in child neurodevelopment. So far, some studies have found a single factor for all EFs [26], and other studies have proposed a two-factor model instead [29,30,31].

Furthermore, studies on children differ from studies on adults in broader processes of defining EFs. For instance, Diamond [32] includes working memory and cognitive flexibility instead of updating and set-shifting, which are more specific processes. Indeed, working memory here refers to a domain-general system that can store and process information simultaneously. It shows a linear increase from ages 4 to 14 and a levelling off between ages 14 and 15 [33]. In contrast, updating is the specific ability to change temporarily stored information in the light of incoming information and is mainly investigated in studies with adults and older school-aged children. Developmental studies have shown that updating increases with age along with upgrading of inhibition efficiency, and stabilizes by the age of 15 years [34]. Cognitive flexibility refers to a tendency to perform in ways that are not fixed or routine, to “think outside the box”, or to adapt to changes in the environment; instead, shifting refers to the ability to switch between conflicting operations or different task sets. Shifting is a more specific dimension than “cognitive flexibility”. However, some authors have pointed out that there is no evidence that cognitive flexibility can be considered a general, coherent construct usable in individual difference research with children [35]. Very often, the term “cognitive flexibility” in developmental studies is actually used with the meaning of “shifting” (e.g., [29]). The development of successful shifting seems to depend on inhibition and working memory. As Garon et al. [36] noted, before children can successfully shift between response sets, they must be able to maintain a response set in working memory and then be able to inhibit the activation of a response set to activate an alternative one. Developmental studies have revealed that shifting improves from age 4 to adolescence, reaching adult-like levels around 15 [37].

Other authors have included different types of inhibition in their definitions of EFs, distinguishing inhibition on a behavioural level (response inhibition or behavioural inhibition) and a cognitive/attention level (interference suppression or interference control), both sharing the need to suppress an action or a thought in order to control impulses and stay focused [32,38]. Studies on their development reveal that, at 4 years, these two inhibition processes are already distinguishable [39]. Improved behavioural inhibition tends to stabilize by the early school years (i.e., from 5 to 8 years; [28]), whereas a sensitive increase in interference control occurs during elementary school and is followed by slower improvement during early adolescence [33].

Furthermore, with increasing age, complex high-order EFs such as planning and problem solving become relevant to be included in the construct of EFs [32]. They develop particularly late in childhood and undergo a final growth spurt during the beginning of adolescence [40,41]. Research on these processes has examined chiefly the development of performance at Tower-like tasks across different age groups and found age effects only for the more complex problems [42].

### 1.1. NC and EFs: Are They Linked?

There are different reasons to expect that EFs and NC are related across development.

In general, the literature frequently reports significant relationships between EFs and different aspects of language skills. Especially during the preschool years, language skills undergo rapid development: vocabulary overgrows, the use of syntactic rules becomes more adult-like, and the ability to use language in narratives improves (e.g., [43,44,45]). At the same time, the preschool years are characterized by a substantial improvement in EFs that are commonly impaired in children with language disorders (e.g., [46]).

The fact that developments in NC emerge in concert with developments in EFs suggests a potential developmental relationship between these abilities. Evidence from imaging studies indicates that these skills depend upon overlapping neural substrates, mainly frontal lobe function, and deficits across these skill sets are observed in adults with traumatic brain injuries [47,48]. However, it is possible to find specific brain regions associated with narrative competence such as temporal poles, the posterior cingulate, and the left superior temporal gyrus [49]. On the other hand, cognitive executive functions are more associated with the bilateral dorsolateral prefrontal cortex [32].

Telling a good story requires the individual to set the goal of linking all of the story elements in a coherent manner, retrieving the appropriate semantic information, syntactic structures, and morphological features that would express the causal links between various story elements, and also indicate the characters’ motivations and reactions, and monitor the narrative while it is being produced. In order to tell a coherent story, children need to set up a hierarchical goal and plan and monitor the organization of the narrative events, and this seems to engage EFs [50]:shifting may be involved in the generation of complete episodes within a narrative discourse, in the selection of informative words, and in the ability to monitor the communicative flow;updating of working memory may be required to generate and understand sentences as well as recall episodic contents for an accurate organization of a story;inhibition processes may be critical for monitoring the production of extraneous comments and derailments while telling a story and for the ability to inhibit the semantic competitors while producing words;planning and more complex EFs may be recruited to the extent of coordinating all the processes involved, as well as for the planning and goal setting of the story (e.g., retelling a narrative containing all of the story elements in the correct sequence [51]).

In the same way, NC development may support the performance on EF tasks. This seems especially plausible on tasks with long and complex instructions and linguistic stimuli to be processed or producing oral responses [52].

However, both cross-sectional and longitudinal studies are inconsistent regarding the association, and potential causal relation, between EFs and NC. For instance [51], in a study on children between 3 and 6 years old, results showed that narrative production is best predicted by high-level EFs, measured with planning and shifting tasks. In contrast, other studies investigating the relationship between these domains in 4–5- and 7–8-years-old Turkish children found that narrative production, especially plot complexity, is related to these EFs only in the older group, not in the younger age band [53]. Moreover, other studies report no association between planning skills and the quality of written narratives in fourth-grade children [54].

A significant relationship between microstructural competence, such as lexical variety and syntax used in narratives, and shifting ability, addressed by the performance at card sorting task, is found in a sample of 47 four- to six-year-old Swedish children. In the same way, EFs accounts for 7% of the variation in syntactic complexity in Turkish-speaking preschoolers [53]. Longitudinal research on school-aged Dutch children reveals that the development of syntactic complexity in narratives between fourth and sixth grade is also predicted by planning and behavioural inhibition in the fourth grade [54]. The relationship between syntactic complexity and inhibitory skills is not found at preschool age in typically developing Swedish children [52].

Research on the role of working memory in narratives appears more consistent. A study on children aged 5 to 8 shows that the ability to update working memory is moderately associated with referential adequacy, the macrostructural competence to introduce and maintain a reference to story characters in narratives [55]. Studies on children aged 8 to 11 reveal that working memory and shifting significantly account for plot complexity variance, another macrostructural NC indicator, in written narratives [56]. Even when controlling for vocabulary, working memory correlated with text generation at the word, sentence, and text level in a sample of 10 years old children [57] and adolescents [58]. According to the authors, it may be involved in translating ideas in the memory into linguistic representation, organizing thoughts into temporally sequenced discourse, and revising text.

In general, studies on narrative writing show that children with higher updating and inhibitory skills produce longer, coherent narratives. The authors [58] explain the involvement of these processes with the need to suppress inappropriate lexical representations, select the relevant ones, and actively hold and update the representations in WM during writing composition. However, some studies on 5- and 6-year-old children with SLI found a significant correlation between narrative retelling skills and working memory, but not with inhibitory processes [59,60].

Furthermore, some studies fail to find a direct relationship between NC and inhibitory and WM updating skills, showing that the influence of these EF domains on NC may totally depend on handwriting skills [61]. Indeed, studies reported that children with poor handwriting skills tend to use the first linguistic expression that occurs to them to frame their ideas without being concerned about shaping the linguistic expression in response to narrative demands or the reader’s needs [62,63,64]. They must devote most or all of their cognitive effort to spelling and handwriting, leaving little resources available for other writing processes. This may limit the amount and quality of text they can generate.

In sum, there is conflicting evidence about the developmental stages at which EFs relates to NC. Inconsistent results suggest that the development of these skills can be heterochronous with ones that are deeply conceptually related and developing on different timescales. Even though they develop across the preschool period, it seems they do not do in lockstep. Some aspects of EFs may develop before others, and the relationship between these aspects and NC may be such that there is specificity in predictive relations over developmental time for microstructural and macrostructural elements [65]. Research with atypically developing populations presenting deficit in both EFs and NC show similar inconsistent results. For instance, in children with a diagnosis of ADHD and language impairment, Fernandez et al. [66] found a significant correlation between macrostructural elements produced in the narration (e.g., episodic structure) and planning skills, but not with phonological working memory. Some studies conducted in children with SLI, instead, found a significant association between plot structure and phonological working memory [59,67].

To date, our understanding of how and when different aspects of NC relate to EFs—or which part of EF they relate to—is limited. Integration of divergent findings has become a necessary and important task. The present study takes up this task using a meta-analytic approach in order to examine and explain the variability across findings. Larger sample approaches may indeed improve our knowledge on the relationship between EFs and NC over developmental time and orient future research on this topic. Currently, to our knowledge, there are no systematic reviews or meta-analyses addressing this issue.

The understanding of how different aspects of NC relate to EFs—or which part of EF they relate to—is also clinically relevant since both the skills predict important life outcomes (i.e., academic and social success) and are trainable [68,69,70,71]. Studies show that children—especially those at risk (e.g., children from backgrounds of poverty, children whose first language is not the one spoken in the country where they live, or children with psychopathological traits)—often exhibit less-well-developed language and executive skills, facing greater risks to academic success than do their typically developing or more privileged classmates [68]. The disadvantages attributed to a lagging NC and EF development increase as children progress through school [71]. Early interventions that support the development of narrative skills in young children have been shown to be effective at promoting NC and academic success at the preschool level (e.g., [72]). Furthermore, these interventions appear to have positive and substantial long-term effects. Evidence on EF training at preschool age also showed that cognitive training to improve these skills early could be effective [69,70].

### 1.2. Aims of the Study

The goals of the present meta-analysis are the following:Determine the overall strength of the relationship between narrative competence (NC) and executive functions (EFs) across childhood and adolescence (3–18 years)Determine *if* the strength of this relationship changes across childhood and *when* it changes across development.Examine potential moderators to understand if the strength of the relation changes:between typically vs. atypically developing children (e.g., attention deficit hyperactivity disorder (ADHD), autism spectrum disorder (ASD), specific language impairment (SLI)).by different EF domains (working memory capacity and updating, behavioural inhibition, interference control, shifting, planning, and problem-solving);by different narrative types (oral vs. written) and levels (micro vs. macrostructural levels).


## 2. Methods

### 2.1. Operational Definitions

We categorized NC based on the characteristics of narratives: written or oral. Both types of narratives included the ability to retell or tell a story in written or oral form. Moreover, we classified measures related to NC by dividing them into micro-structural and macro-structural competence. Micro-structural components were collapsed into one dimension, including lexical (e.g., number and variety of words produced) and syntactic skills (e.g., indices of number and type of utterance and subordinate sentences produced; the mean length of utterance) in narration. Macro-structural components were collapsed into one dimension, including the richness of content of the narrative (e.g., the amount of information reported in the narrative), the presence of the key passages in the story (e.g., the ability to structure a coherent story), and the cohesion of the story (e.g., anaphoric use of the article and correct referencing across the narration).

Executive domains were differentiated according to which primary executive process the tasks assessed, based on the EFs assessment literature [32,38,42,73]. For instance, tasks requiring keeping in mind and actively manipulating auditory or visual information (e.g., backward digit; word or spatial span tasks) were coded as working memory capacity measures. These were distinguished from tasks that mainly required updating of working memory (e.g., n-back), defined as “the ability to monitor and code incoming information, and to update the content of memory by replacing old items with newer, more relevant, information” ([74] p. 428). Forward span-like tests were considered to measure short-term memory since they did not require working memory processes [75]; therefore, we did not include them in the meta-analysis.

We considered those tests that required children to suppress a dominant but inappropriate response or to prevent impulsive motor response (e.g., knock and tap task; go/no-go; Head Toes Knees Shoulders) as a measure of “behavioural inhibition” [38]. Instead, tasks requiring the ability to prevent interference due to resource or stimulus competition and filter out irrelevant information within the stimuli that contain both relevant and distracting information (e.g., Stroop-like, local to global and Flanker paradigm) were categorized as “interference control” task [38]. 

We categorized tests requiring shifting among different response sets and flexibly adjusting the response according to new rules (e.g., verbal fluency, five-point test, Trail Making Test and Wisconsin Change Card Sort) as measures of “shifting”.

We classified tests that required the ordering of events mentally in advance and planning of actions [76], such as Tower-like tasks or non-narrative sequences, as measures of planning abilities.

If a study collapsed different tasks in a single general dimension, we included it as a general measure of EF for the purpose of the main analysis (e.g., [31,46]. However, in such cases, we could not be able to discern between the various EF domains implied. For this reason, we could not consider such outcomes for the analysis of moderation by EF domains.

### 2.2. Search Strategy

In accordance with the PRISMA statement [77] we used a systematic search strategy to find the pertinent studies. Using different combinations of the terms “executive functions”, “narrative”, and their synonyms (see Appendix B for the detailed search strings), we searched on PubMed, PsycINFO, Linguistics and Language Behaviors, Proquest Dissertations and Theses Global, e-thesis online service (Ethos), DART-Europe E-theses Portal to identify all potential journal articles, unpublished studies, and doctoral dissertations that reported data on the relationship between EFs and NC in children and adolescents. This is the first meta-analysis on narrative competence and executive function in children and adolescents. Despite our extensive search of the grey literature, we found only a small amount of unpublished studies (overall, 5 studies and 46 different effect sizes). Preliminary analyses ruled out the presence of publication bias: the size of the relationship was similar in the published and unpublished studies. Therefore, we also included these studies in the main analysis.

After excluding duplicates, 885 records remained. The first author screened all of them based on title and abstract and according to inclusion and exclusion criteria. As a secondary search, the references of the selected studies (n = 15), in addition to relevant systematic reviews, were checked to find other eligible studies. The full text of the identified papers was reviewed by the first author and EB. Disagreements were solved through discussion. The agreement rate between the two raters was high (81%). Finally, as shown in the flow chart, we identified 25 articles (30 studies) with 287 effects that were eligible for the present meta-analytic review. Details concerning the literature search method and criteria for inclusion and exclusion of studies are shown in Figure 1.

### 2.3. Inclusion Criteria

The included studies had to meet the following criteria:at least one performance-based test related to EFs and one related to the micro- or macrostructural level of NC;correlational study with a cross-sectional or longitudinal design;monolingual participants aged between 3 and 18 years old;paper is written in English, Italian, or Spanish.

### 2.4. Exclusion Criteria

We excluded all the studies where participants were bilingual and older than 18.

All outcomes were based on correlations between one or more EF and NC tasks. Where available, we included correlation with accuracy and reaction times on EF tasks. We did not accept measures of EF aspects collected through teacher and parent reports (e.g., BRIEF) because these measures seem to capture different aspects from tasks [78]. At the same time, we did not accept measures of narrative comprehension measured through questions. The only kind of NC tasks included required the child to produce a personal story or to retell a story they heard, in oral or written form. We included the studies only if they reported at least one score of a neurocognitive EF measure and at least one micro- or macrostructural competence score for an NC task.

### 2.5. Coding

During the coding phase, the first author coded each record according to a predefined coding schema, collecting information about bibliographic information (i.e., title, author(s), and year of publication), sample characteristics (i.e., sample size, mean age and standard deviation of each group, clinical risk status of the sample), characteristics of the narrative tasks (i.e., written versus oral form; microstructural versus macrostructural level) and the kind of EF measure (i.e., working memory capacity, updating of working memory, behavioural inhibition, interference control, shifting, and planning) and the correlation indices between the NC and EF tasks.

All the correlation indices between the tasks were included if there were two or more eligible NC and EF measures. We applied the same procedure when multiple groups were suitable for the aims of the meta-analysis, like typically and atypically developing children in the same study (i.e., [60,79,80]) or preschoolers and school-aged children (i.e., [53,55]).

### 2.6. Meta-Analytic Procedures

We used R version 4.1.2 [81], RStudio version 1.4.1103 [82], and the Metafor package [83,84] to conduct the analyses. R code and data are openly available in Appendix A.

Pearson product-moment correlation was used as the effect size to examine the relationship between NC and EFs. The magnitude of the correlation was interpreted using Cohen’s [85] conventions:*r* ≈ 0.10 [*z* ≈ 0.10]: small effect;*r* ≈ 0.30 [*z* ≈ 0.31]: moderate effect;*r* ≈ 0.50 [*z* ≈ 0.54]: large effect.

Since correlations are restricted in their range (i.e., they can take values between −1 and 1), it can introduce bias when we estimate the standard error for studies with small sample size. Thus, the correlation coefficients collected from the selected studies were transformed into Fisher’s *z*. This transformation entails using the natural logarithm function to remove the range restriction and ensure that the sampling distribution is approximately normal. Fisher’s *z* and the standard error of Fisher’s *z* were calculated directly in R using the cor and log functions.

A positive *z* value reflected a positive association between NC and EFs, while a negative effect indicated that when the EF competence increased, NC decreased. We computed Z Fisher transformation using Olkin and Finn’s [86] formula. The summary statistics required for each outcome were the number of participants and the correlation coefficients between NC and EF measures. For one study based on regression analysis (i.e., [51]), the correlation coefficient was converted from the β coefficient, according to Peterson and Brown’s [87] procedure.

As discussed, many studies in the dataset reported several correlated relevant outcomes, and some studies comprised multiple groups of individuals (e.g., with typical and atypical development). This caused dependencies in the data. So far, several solutions have been introduced to avoid dependency [84,88]: analysing the outcomes as if they were independent (i.e., ignoring the dependency), averaging the dependent outcomes into a single effect size, selecting only one outcome for each study, and multilevel meta-analysis. Ignoring the dependency might bias the results; averaging or eliminating effect sizes, on the other hand, would decrease the power of the analysis and limit the research questions that we could ask, as we would not be able to compare moderation effects by EF and NC domains. We therefore conducted a three-level meta-analytic analysis, following Assink and Wibbelink [84]. The meta-analytic model considered three different sources of variance: the participants at level 1, the outcomes at level 2, and the studies at level 3.

We used the rma.mv function of the Metafor package and set the tdist parameter as TRUE. Therefore, we based the test statistics and confidence intervals on the t distribution, applied the Knapp and Hartung [89] adjustment, and used the Restricted Maximum Likelihood estimation method (REML) for estimating the parameters. Tau^2^, the Q-test for heterogeneity [90] and the I^2^ statistic were reported.

Studentized residuals and Cook’s distances were used to examine whether studies may be outliers or influential in the model context. Studies with a Studentized residual larger than the 100 × [1 − 0.05/(2 × k)] th percentile of a standard normal distribution were considered potential outliers (i.e., using a Bonferroni correction with two-sided alpha = 0.05 for k studies included in the meta-analysis). Studies with a Cook’s distance larger than the median plus six times the interquartile range of the Cook’s distances were considered influential.

## 3. Results

### 3.1. Selected Studies

Thirty studies were eligible for inclusion, for a total of 287 different outcomes, with 3250 participants with typical development (M_age_ = 8.18) and 346 participants (M_age_ = 8.02) with atypical development (i.e., diagnosis of learning disorder, autism spectrum disorder, language impairment, deafness).

### 3.2. Inspection for Publication Bias

We explored the funnel plot to investigate potential publication bias and checked for differences in effect sizes between published and unpublished studies. The Egger’s regression test, using the standard error of the observed outcomes as a moderator, was used to check for funnel plot asymmetry. The funnel plot is presented in Appendix A.

No evidence of publication bias emerged, (Egger’s *t* = 1.116, *p* = 0.266). A visual inspection showed that only a few studies fall outside the pseudo-confidence interval’s triangular region. Next, we compared the effect sizes of published and unpublished studies, as higher effects for published studies might be an important indication of publication bias. We could locate only five unpublished studies, with a total of 46 different outcomes. No evidence of publication bias emerged, *F*(1, 285) = 0.96, *p* = 0.325. On the contrary, the size of the effect was slightly bigger for the five unpublished studies than for the published studies: for the unpublished studies the effect was *z* = 0.283, SE = 0.041, 95% CI = (0.199, 0.367) and for the published studies the effect was *z* = 0.233, SE = 0.020, 95% CI = (0.193, 0.273). Since this difference was negligible, we decided to include the five unpublished studies in the main analysis.

Subsequent analysis indicated that the size of the effect was related neither to the year of publication of the study, *F*(1, 285) = 0.187, *p* = 0.665, nor to languages spoken by the sample of participants involved in the studies, *F*(7, 296) = 0.193, *p* = 0.986. Moreover, a sample size moderator analysis was performed, which resulted in a non-significant effect (*p* = 0.109), suggesting that differences in sample size are not a source of the heterogeneity of the results.

An examination of the Studentized residuals revealed that one study [91] had a value larger than ±3.7537 and may be a potential outlier in the context of this model. According to Cook’s distances, four studies [79,92,93,94] could be overly influential.

### 3.3. Research Question 1: Exploring the Overall Association between EFs and NC

A total of k = 287 effects were included in the analysis. The observed Fisher r-to-z-transformed correlation coefficients ranged from −0.0601 to 1.2111), with the total estimates being positive. The estimated average Fisher r-to-z-transformed correlation coefficient based on the random-effects model was *z* = 0.241, *r* = 0.236, (95% CI: 0.2053 to 0.2776). Therefore, the average outcome differed significantly from zero (*t* = 13.134, *p* < 0.0001), indicating a positive, small association between EFs and NC over development. According to the Q-test, the true outcomes appear heterogeneous (Q_(286)_ = 597.25, *p* < 0.0001. The estimated variance components were tau^2^_(level 3)_ = 0.005 and tau^2^_(level 2)_ = 0.006. This means that I^2^_(level 3)_ = 22.95% of the total variation can be attributed to between-study and I^2^_(level 2)_ = 29.95% to within-study heterogeneity. We found that the three-level model provided a significantly better fit compared to a two-level model, with level 3 constrained to zero (χ^2^ = 33.39, *p* < 0.001).

The 75% rule (Hunter and Schmidt, 1990 [95]) suggests that we should inspect heterogeneity if <75% of the total amount of variance can be attributed to within-study sampling variance. Therefore, we proceeded to investigate potential moderators, following the research questions outlined above.

### 3.4. Research Question 2: Exploring If and When the Association between EFs and NC Changes over Development

We investigated the impact of age on the relationship between EFs and NC through meta-regression to understand if and when the relationship between NC and EFs changes over time (see Table 1). The mean age of the sample ranged between 4 and 15 years and significantly influenced the effect size so that as age increases, the overall effect size decreases, *F*(1, 265) = 6.744, *p* = 0.009.

The unstandardized regression coefficient and significance for the slope are reported in Table 1, which indicates the impact of each unitary change (i.e., one year) in the moderator on the effect size of the relationship between EFs and NC.

Looking at the trend in effect size over development (see Figure 2), the relationship’s turning point appears to be around 7–8 years old. Thus, we performed moderation analysis by dividing the sample into two-time windows (i.e., mean age < 7 years; mean age > 8 years). Results show that this variable significantly impact on the effect size, so that after 7 years old the magnitude of the relationship between EFs and NC decreases from *z* = 0.274 to *z* = 0.212, *F*(1, 265) = 3.908, *p* = 0.049. According to these results, we decided to conduct separate meta-analyses to investigate the influence of potential moderators in these two developmental windows (4–7 years; 8–15 years, see Table 2).

Table 3 and Table 4 summarized the characteristics of the studies included in the first and second meta-analysis, respectively. In particular, in Table 3 we reported the correlations between EFs and NC of participants aged 4–7 years old; in Table 4, we reported the correlations between EFs and NC of participants aged 8–15.

### 3.5. Research Question 3: Potential Moderators of the Relationship between EFs and NC before and after 7 Years Old

As previously mentioned, Table 2 shows a summary of the impact of the following moderators on the relationship between EFs and NC in the two developmental windows considered.

*Typically vs. atypically developing population.* We categorized the sample in typically developing and atypically developing participants based on the presence of a diagnosis (i.e., deafness, SLI, learning disorders, ADHD, and ASD). The studies involving children younger than 7 years old (n = 795) indicated that the effect sizes differed between the groups, *F*(1, 83) = 4.400, *p* = 0.039. The association between EFs and NC was almost twice in atypically developing children (*z* = 0.436) than in typical peers (*z* = 0.249), unless both effects are significant.Conversely, in the subsample of studies involving children older than 8 years old (n = 2615), the analysis indicated that the effect size was the same for typically (*z* = 0.211) and atypically (*z* = 0.196) developing populations, *F*(1, 180) = 0.132, *p* = 0.715.The number of studies involving atypically developing populations of children, however, was relatively small in both subsamples: we found only four studies with a total of eight different effects and 143 atypically developing children younger than 7 years old; and only five studies with a total of seventy-six different effects and 203 atypically developing children older than 8 years old.*EF domains.* Looking at EFs, we investigated if, before and after 7 years old, effect size differs on the type of EF domains taken into consideration (i.e., interference control, behavioural inhibition, working memory capacity, updating of working memory, shifting, planning). Results showed that before 7 years, the effect size did not statistically differ based on the type of EF domains, *F*(5, 77) = 2.069, *p* = 0.109. At this stage, EF domains are equally significantly associated with NC. However, in the subsample of studies involving participants older than 8 years old, variance in the effect size was significantly explained by EF domains, *F*(5, 162) = 3.399, *p* = 0.006. In line with the age effect previously discovered, the relationship between NC and the majority of the EF processes decreased, with the exception of behavioural inhibition. The effect size of the association between behavioural inhibition and more general NC was larger than those observed in younger children.

Additionally, the association of shifting and planning with NC remain significant in older children, even if it is lower. As regards working memory dimension, the measures addressing its capacity remains similarly associated with NC, whereas those addressing updating processes decreased significantly in older children.

*Narrative Competence.* Looking at the characteristics of NC, we next compared studies on children before and after 7 years old, analysing if micro versus macrostructural levels of narratives moderated the effect size of the relationship between EFs and NC. Results referring to studies on participants younger than 7 years old indicated that the effect size was higher for macrostructural (*z* = 0.329) than microstructural (*z* = 0.208) competences, *F*(1, 75) = 12.23, *p* < 0.001, unless both the effects were significant (*p* < 0.001). After 7 years old, however, no significant difference emerged for the comparison between micro and macrostructural aspects, *F*(1, 180) = 0.074, *p* = 0.784.

Next, we questioned if, in the subsample of studies with children older than 8 years old, the relationship between EFs and NC differed based on the type of narrative tasks (i.e., written versus oral form). Results indicated that the type of narrative task did not explain variance in the effect size, *F*(1, 180) = 1.36, *p* = 0.243.

## 4. Discussion

EFs and NC are two widely investigated dimensions of human cognitive development, but our understanding of their relationship is limited. For instance, we do not know if these dimensions are related over time or if this relationship changes across development. We do not know much about this relationship, especially in atypically developing children and adolescents, although we know that these areas are usually impaired in such populations. In general, few studies have investigated this relationship. Mostly, these studies involved small samples, used a cross-sectional design, and produced mixed results. The aim of this meta-analysis is not to answer these questions according to the studies published so far. It intends to raise some points that can guide future research on these topics, such as which age range needs further consideration by scientists. We claim, as of right now, that more studies in general—and specifically more longitudinal studies—are needed to shed light on the relationship between these dimensions over time in typical and atypically developing individuals.

The first purpose of the present meta-analysis was to establish if, overall, EFs and NC are transversally—not longitudinally—associated.

As expected, the collected studies showed great heterogeneity within and between themselves. However, the multilevel meta-analysis showed that—overall—a positive but small relationship between EFs and NC exists (*r* = 0.236). It means that the studies selected provide evidence that—in general—individuals who performed well at EF tasks are also good narrators and vice versa. The result obtained reflects the high variability between the studies included. Nine studies reported an average effect size below 0.20, but most reported moderate (0.30–0.49) effect sizes. Inspection for publication bias revealed that the results obtained are similar in the published and unpublished literature, so the probability of overestimating the magnitude of this relationship is remote.

The second purpose was to examine if the relationship between EFs and NC changes over time and at which point it starts to change significantly. In order to fulfil this aim, we considered the mean age of participants in the studies. Results showed that the relationship between EFs and NC changes over time and decreases over development. The plot of the association between NC and EFs across development (Figure 2) showed that the transversal association increases during the preschool years, when both NC and EFs dramatically develop, peaking in the early elementary school years and then starting to decrease significantly after 7 years old.

Different factors might explain the turning point we can observe at this age.

We speculate that a key role might be played by literacy acquisition to which the early years of elementary schools are dedicated. During these years, children develop effective decoding skills [98]. Specifically, children speaking languages with shallow syllabic complexity and orthographic depth (e.g., Italian, Spanish, German, Greek) become accurate and fluent in foundation reading before the end of the first school year. In contrast, children speaking languages characterized by deep orthographies (English, French, Danish, and Portuguese)—the majority of children involved in the studies selected for this work belong to this group—become fluent at nearly 8 years old [98].

Research on the development of reading and writing suggests that the development of these skills is deeply interrelated and that, especially during elementary school years, reading contributes significantly to the quality of narrative composition [12,15], especially from a macrostructural point of view (i.e., better structured and cohesive narrations). It is possible that, after literacy acquisition, the role of EFs in narrative production is downgraded by other factors that contribute to NC development, such as reading skills. Of course, this is a speculative interpretative hypothesis. To our knowledge, there are no studies that have taken into consideration the role of both EFs and reading skills on the development of NC.

Changes in exposure to narratives could also explain the decrease in the association between EFs and NC. The amount of this exposure may play a role in the development of NC and downgrade the association between EFs and NC. It is true that narratives are cross-culturally used in childrearing systems, and children are exposed to them from very early in life to a greater or lesser extent. However, during preschool and the first years of elementary school, children are exposed to narratives and narration is widely used as an educational strategy in school. Narratives create a pleasant and creative learning environment and a more general constructive and enjoyable atmosphere for the children [99]. Moreover, the use of narrative in education attracts the interest of the children and assists in the better understanding of the information obtained through this. Often, story grammar becomes part of the school curriculum and children are taught to become good narrators, so it is possible that when the development of good NC becomes formal learning, NC may progressively be less associated with or dependent on EFs. 

It seems that the two dimensions are more associated early in childhood, the period in which EFs and NC—taken singly—dramatically undergo rapid and qualitative changes [10,100]. We have discussed the possibility that EFs may become less relevant for supporting NC over the course of development, but it is also possible that NC supports EF development across time, becoming less essential by nearly 8 years old. There is evidence that language skills support EF development, especially across preschool age, and narrative language could be considered a “naturalistic” way to investigate children’s language in connected speech [1]. Therefore, it is possible that the practice of constructing causally coherent true narratives could help children in initiating and regulating behaviour—as demonstrated in language research [101,102]—and that narrative language may have a mediating role in EF performance, as there is evidence that language skills have this role in both deaf and hearing children [103]. 

However, it is still possible that increasing cognitive demands associated with the transition to elementary school and the development of other competencies play a more significant role than NC in the development and reorganization of EFs. The role of NC—and language—in EF development can be progressively nuanced by the other increasing competencies in this period, which could be responsible for the decrease observed in their association. It should be pointed out that the argument that the magnitude of the relationship between NC and EFs seems to decrease over time applies only to the transversal relationship between them. One competence may relate longitudinally with the other and vice versa. For instance, NC and EFs may be weakly related at 9 years old, but EFs at 5 years old is significantly associated with NC observed at 9 years old. However, there is insufficient data in the literature to answer this question with a meta-analysis.

The third purpose of this work was to try to understand some moderators responsible for the heterogeneity observed between studies in the magnitude of the association between EFs and NC. Since the magnitude of the transversal relationship between EFs and NC changes over time, we analysed the role of these moderators in two different time windows: before and after 7 years old.

We found that, before 7 years, the association between EFs and NC is stronger in children with atypical development, such as ASD, ADHD or SLI. However, later in development, the strength of the association fades. After 7 years, results suggest the strength of the association appears similar in typical and atypical development unless only the latter is statistically significant.

As mentioned above, NC and EFs are skills that predict important life outcomes and are trainable [68,69]. They are frequently impaired in children with ASD, ADHD or SLI [46,104], and our results may suggest that in such populations the impairment on EFs could somewhat impair NC, or vice versa, between 3 and 7 years of age. In the literature, several training programs aimed at improving EFs or NC have been described (e.g., [72,105], showing promising results in preschoolers [70,71]. There is also evidence that the training effects are higher in children with developmental risks or psychopathological traits [70,106,107].

Establishing if two dimensions are associated across development is the first necessary step to hypothesize that training one could foster the development of the other. Currently, research aiming to study the effectiveness of EF or NC training did not take into consideration possible far transfer effects on them. In the same way, there are no studies that implement integrated interventions targeting both NC and EF or studies that verify their effectiveness.

The results of this meta-analysis could be read as a first step towards research on integrated interventions and plans to verify the effectiveness of single EF intervention on NC and vice versa. Based on our results, we could propose some speculative hypotheses related to the fact that—if a far transfer between NC and EFs is possible—the chance to observe it on the non-directly trained skills would reduce after 7 years. Following this reasoning, according to our results, only training programs aimed at improving single specific competencies might be effective in older children showing impairments both in NC and EFs. This is consistent with research that unanimously agrees that intervention is likely more effective and pervasive when provided earlier in life rather than later [108].

Moderation analyses also explained part of the heterogeneity in the effect size between and within different studies depending on different EF domains and NC levels assessed.

We found that before 7 years the association between EFs and NC is stronger if we considered the macrostructural level of NC, which includes several important story characteristics such as the quantity of information, story structure, and cohesion. This is not unsurprising, as in the transition from preschool- to school-age this competence shows a remarkable increase (e.g., [20,21]). For instance, analysing the stories produced by children aged 4 to 8, Schneider et al., (2006) [97] showed a significant increase in the quantity of relevant information included in the narrations as children’s age increased. In addition, as children grow and develop their NC, they gradually move from non-goal-directed sequences toward complete episodes. From preschool to elementary school, children go from producing stories that include few causal connections between events to being able to conceive an overall plot with most of the story grammar elements and following a logical progression of events in their stories, which make them appear more cohesive and well structured. It is possible that EFs play a significant role in this progress and that this progress may support EF development.

It seems that, later in development, the strength of the association between EFs and macrostructural NC fades. After 7 years, results suggest the strength of the association appears similar at the microstructural and macrostructural levels. However, children’s ability to tell stories continues to develop during primary and secondary school. Older children indeed include more events than do younger ones [7]; they correctly use a broader range of conjunctions [109] and more advanced anaphoric strategies (e.g., pronouns were used to maintain a reference to characters, whereas nominals were used to switch a reference) that make the stories appear more cohesive. Additionally, EFs show an increase in late childhood and adolescence, but its development may be less involved in NC and vice versa. As argued before, a more significant role in NC increase at this time may be played by reading skills consolidation or other competencies.

Heterogeneity in the effect size seemingly cannot be explained by the narrative form (oral vs. written) used in the articles collected. This is consistent with results found by Bigozzi and Vettori [16], who showed that, in the transition from oral to written code, typically developing children who master writing preserve their oral narrative skills. There is evidence that difficulties in written over oral narrative form may be observed in atypically developing children who struggle with handwriting. Unfortunately, our sample size was not adequate to investigate the interaction of the two moderators (i.e., population and narrative form) in the subgroup analysis. In the subgroup of studies involved in the second meta-analysis (children older than 8 years), atypically developing children represent only 8% of the sample.

As regards EF domains, we found that the strength of the association between EFs and NC appears similar for different EF domains before 7 years old. After 7 years, results showed a general decrease in the strength of the relation, even if some differences from medium overall effect size emerge by different EF domains. 

More specifically, in preschoolers and first and second graders, the contribution of EFs to NC appears statistically equal across EF domains. This could be because, at this age, EFs tend to be more related and less differentiated from each other [26,29,30,31], so any attempts to connect the various tasks to one distinct EF domain at this age may be artificial. For this reason, specific patterns between EF domains and NC could be challenging to observe in this time window. Additionally, a technical consideration may explain the absence of evidence. Studies included in the first meta-analysis showed substantial between-study heterogeneity within the EF domains, which decreased the pooled effect’s precision (i.e., increased the standard error). Yet, when the EF domains effect estimates are imprecise, their confidence intervals will have a large overlap, as in some of our cases (e.g., working memory updating CI index: 0.057, 0.632). Consequentially, this might make it harder to find a significant difference between subgroups—even if this difference could exist.

Specific patterns in the relationship with NC may emerge after 7 years old, when EF domains are more differentiated and distinguishable [28].

In general, the contribution of all EF domains to NC seems to decrease after 7 years, with the notable exception of behavioural inhibition. This domain refers to the ability to suppress a dominant but inappropriate response or prevent impulsive motor response, according to Nigg’s definition [38]. Together with interference control, behavioural inhibition may be critically involved over development to monitor the production of extraneous comments and derailments while telling a story or inhibit semantic competitors while producing words. NC may also be involved in inhibition tasks. Narrative language may indeed be used to exert control over attention and inhibit inadequate response and interferent representation. 

As with inhibition, working memory capacity, shifting, and planning also appear to be involved in NC at this age. Working memory capacity could be required to keep in mind ideas before translating them into linguistic representations, as well as to recall episodic contents for an accurate organization of temporal sequences in the story. Shifting could be required in the generation of complete episodes and in the ability to monitor the communicative flow. Instead, planning may play a coordinating role in story organization, e.g., putting all the story elements in the correct sequence [51]. These results are in line with studies reporting that working memory, shifting, and planning are correlated with text generation in older children [57,61] and adolescents [58]. Other domains seem to be significantly less associated in this period with NC than in the previous time window, such as updating of working memory. This is consistent with previous findings in Swedish [52] and Canadian preschoolers [55].

### Study Limitations

Finally, we would like to discuss some limits of the present work. As claimed above, the current meta-analysis cannot respond definitively to some questions about the relationship between NC and EFs because of its limits. The first limit is related to the fact that few studies investigate this relationship with a longitudinal design. Therefore, even if our results clearly show that a relationship between NC and EFs is definitively positive, we know it is just transversal. We cannot say something about how and if these dimensions are related longitudinally across time if there is one point at which one predicts the other and vice versa because there is not enough research addressing this issue. Future research should investigate if these domains are predictive of each other and establish the direction(s) of their development. A second limitation concerns the time variable used in this meta-analysis to answer the question of whether the relationship changes over time: the mean age of participants. Some studies included in the present meta-analysis involved participants of a large range of ages (e.g., 7–12; 7–14, see Table 4), so it was hard to classify the studies by age stage (e.g., preschoolers; school-aged; adolescents). We preferred not to exclude these studies from the analysis and chose to consider the mean age of the participants collecting—where available—the effect size adjusted for the effect of age. The time effect is one of the most interesting issues for a developmental psychologist. Even if the praxis to analyse the impact of time/age over a phenomenon in meta-analytic developmental psychology research is consolidated, it should be kept in mind that using aggregate information—such as the mean age of participants—may produce ecologically biased results [110,111]. Therefore, any conclusion around the relationship between EFs and NC changes should be taken cautiously and considered just orientational. Aggregating data suggest that a turning point in this relationship occurs at around 7–8 years old, but studies covering this age range also include 6- and 9-years-old participants. Furthermore, studies covering this age range in the sample of articles selected from the meta-analysis are few (k = 4). Meta-analytic research led to summarizing results from different studies, which potentially may offer a comprehensive picture of a phenomenon. In this case, we can see that the relationship between EFs and NC seems to decrease over time, even if we cannot be sure of the exact time point at which it starts to drop, but it seems that it takes place around the first three grades of elementary school. Future studies should examine this period in more detail than preschool.

Another limitation concerns the intrinsic multidimensionality and complexity of EF construct examined and the large variety of instruments used to capture the construct across development. We based the instruments’ classification on the scientific literature [32,38,42,73] in order to clarify which task assesses which specific component, but we are aware of the “task impurity problem”, a phenomenon in which one task assesses various EFs components beyond the one it aims to evaluate, which is frequently in young children. So far, we invite the reader to take cautiously into consideration findings about the specific pattern of relationships between various EF domains and NC since this may depend on the classification we used.

Finally, the last limitation we mention is that NC and EFs are two dimensions that, in real life, are related to many other dimensions of human development that could mediate or explain their relationship. One of these is theory of mind, which is associated with both dimensions [60,112]. In certain circumstances, speculatively, these variables might be responsible for the presence or the lack of association between EFs and NC across the studies. Studies included in this meta-analysis consider the account of potentially confounding variables (e.g., age) on the correlation between EFs and NC, to various degrees and differently. They used to control their effects by reporting partialized correlation coefficients of the relationship between EFs and NC. Unless this operation is fundamental to provide a reliable measure of the association between EFs and NC, it increases the between-study heterogeneity. For this reason, another limitation in interpreting our results is that we cannot be sure that this relationship is direct. Further investigations are necessary for this scope.

## 5. Conclusions

In conclusion, despite these limitations, this work suggests that, over time, the domains of EF and NC are associated and may depend on each other. This seems to be especially true in young, atypically developing children and for macrostructural elements of NC. However, in general, the relationship between EFs and NC that is stronger in early childhood is bound to decrease over development. Since these competencies are usually impaired in children with atypical development, but they can be effectively trainable, we stress that good practice might be to introduce small group interventions to support one or both competencies at the end of preschool and in the first two grades, i.e., at the time EFs and NC appear more related. 

Furthermore, the results provided in this meta-analysis and their limitations suggests some orientational consideration for future research:Previous research has focused more on these domains taken singly than on their relationship. However, to understand human development and support it with effective intervention, we should also focus on connecting its parts. NC and EFs are promising domains because they predict many life outcomes and seem trainable. We should know much about their relationship, especially in atypically developing people and in longitudinal ways. This is to understand when and how it is better to intervene to be effective.Previous research on EFs and NC focused mainly on two age bands (i.e., 3–6 and 9–12) and considered large age ranges. This makes it hard to understand the development of the relationship between EFs and NC across time. Even if results provided by single studies are frequently controlled by age differences, it would be insightful to observe the correlation in more homogeneous age groups. Furthermore, since the strength of the relationship seems to decrease over time, and a turning point in this sense may be represented by the first two grades of elementary school, studies focused on this particular time window—which is been more neglected—should be encouraged to better understand what happens at this specific stage and if we can use it to support child development.

## Figures and Tables

**Figure 1 children-10-01391-f001:**
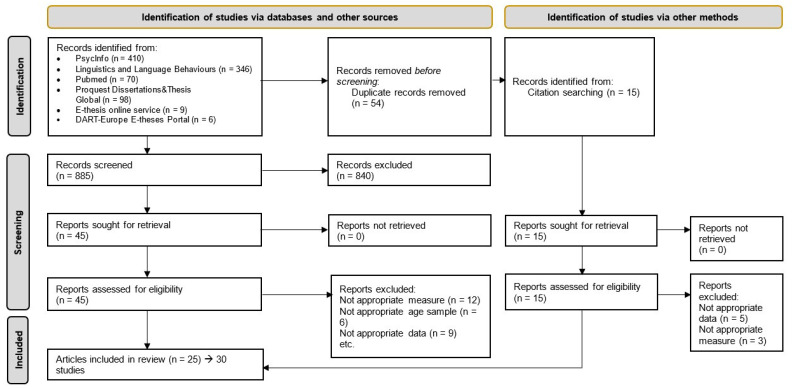
Prisma Diagram. Source: [77]. For more information, visit: http://www.prisma-statement.org/ (accessed on 19 April 2022).

**Figure 2 children-10-01391-f002:**
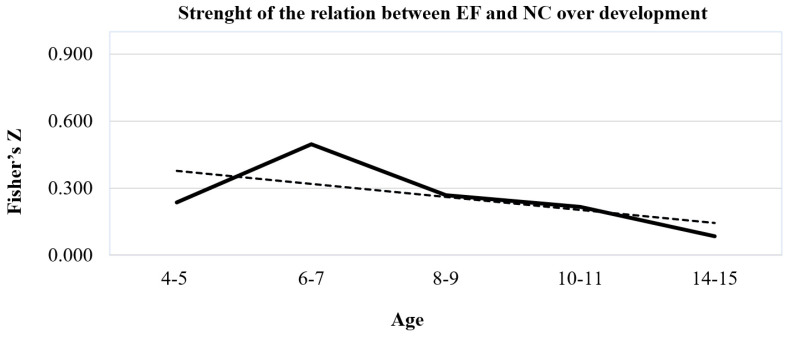
The relationship between EFs and NC over development. Note: The solid line represents the trend of Fisher’s Z coefficient over time. Point of the solid line are averaged effect size of the relationship between EFs and NC in the five time-intervals considered. The dotted line is the trend line of the relationship between NC and EFs over time. The angular coefficient of the dotted line is negative, indicating that the association between NC and EFs decreases over time.

**Table 1 children-10-01391-t001:** Age effect on the relationship between EFs and NC.

Effect	No. Outcomes	No. Studies	No. Participants	Estimated *z*	SE	95% CI		*p*-Value
Children’s age (years)	267	29	3410	−0.014	0.005	−0.025	−0.003	0.009
Developmental time windows								
*Before 7 years*	85	13	795	0.274	0.029	0.216	0.333	<0.001
*After 7 years*	182	16	2615	0.212	0.021	0.170	0.254	<0.001

Note: Italic text indicates the levels of the categorical variables.

**Table 2 children-10-01391-t002:** Moderators of the relationship between NC and EFs before and after literacy acquisition.

Effect	No. Outcomes	No. Studies	No. Participants	Estimated *z*	SE	95% CI		*p*-Value
*4–7 years*: Population								
*Typically* *developing*	77	9	652	0.248	0.230	0.202	0.294	<0.001
*Atypically* *developing*	8	4	143	0.436	0.086	0.264	0.607	<0.001
*8–18 years:* Population								
*Typically* *developing*	106	11	2412	0.221	0.026	0.169	0.273	<0.001
*Atypically* *developing*	76	5	203	0.199	0.040	0.119	0.279	<0.001
*4–7 years*: EF Domain								
*Working memory capacity*	36	7	459	0.259	0.035	0.188	0.330	<0.001
*Working memory updating*	2	1	37	0.344	0.144	0.057	0.632	0.019
*Interference control*	8	2	63	0.309	0.074	0.160	0.458	<0.001
*Behavioural Inhibition*	18	5	185	0.153	0.049	0.055	0.251	0.002
*Shifting*	12	4	211	0.292	0.054	0.183	0.400	<0.001
*Planning*	7	2	122	0.372	0.075	0.222	0.522	<0.001
*8–18 years:* EF Domain								
*Working memory capacity*	39	11	2248	0.232	0.032	0.168	0.297	<0.001
*Working memory updating*	12	1	40	0.135	0.087	−0.036	0.307	0.120
*Interference control*	52	4	1295	0.228	0.044	0.139	0.317	<0.001
*Behavioural Inhibition*	14	4	269	0.292	0.048	0.197	0.387	<0.001
*Shifting*	30	6	339	0.205	0.043	0.119	0.291	<0.001
*Planning*	17	4	177	0.204	0.052	0.101	0.307	<0.001
*8–18 years:* Narrative form								
*Oral*	86	6	266	0.252	0.044	0.165	0.340	<0.001
*Written*	96	10	2349	0.200	0.026	0.148	0.252	<0.001
*4–7 years*: Narrative Competence								
*Micro-structural*	45	8	578	0.209	0.023	0.163	0.0255	<0.001
*Macro-structural*	32	8	527	0.329	0.025	0.278	0.380	<0.001
*8–18 years:* Narrative Competence								
*Micro-structural*	105	12	2476	0.213	0.024	0.164	0.261	<0.001
*Macro-structural*	77	14	1208	0.216	0.026	0.164	0.268	<0.001

Note: Italic text indicates the levels of the categorical variables.

**Table 3 children-10-01391-t003:** Studies including participants aged 4–7 years old.

References	Location	Clinical Risk Status of the Sample	Mean Age (Years)	Age Range	EF Domain	EF Task	Narrative Form	Narrative Competence	NC Indicator	Fisher’s Z, [95% CI]	SE
Balaban et al., 2020 [53]	Turkie	Typically developing (n = 18)	4.42	4–5	Behavioural Inhibition	Emotional Stroop Task	Oral	Macro-structural	Story Content–plot complexity	0.2554 [−0.2506, 0.7615]	0.2583
						Emotional Stroop Task		Micro-structural	Morphosyntactic Complexity	0.4847 [−0.0214, 0.9908]	0.2583
Dodwell and Bavin, 2008 [59]	Australia	Specific Language Impairment (n = 16)	6.70	6–7	Working Memory capacity	Digit Span	Oral	Macro-structural	Information	0.182 [0.3616, 0.7256]	0.2773
					Working Memory capacity	Word Span			Information	0.3205 [0.2231, 0.8641]	0.2773
					Working Memory capacity	Recalling Sentences			Information	0.4059 [0.1377, 0.9495]	0.2773
Duinmeijer et al., 2012 [67]	The Netherlands	Specific Language Impairment (n = 34)	7.35	6–9	Working Memory capacity	Digit Span	Oral	Micro-structural	Mean Length of Utterance	0.6416 [0.2896, 0.9936]	0.1797
Friend and Phoenix-Bates, 2014 [65]	USA	Typically developing (n = 38)	5.00	4–5	Shifting	ANT-executive attention subtest	Oral	-	Story content, lexicon and syntax	0.2693 [−0.062, 0.6006]	0.1691
					Shifting	ANT-executive attention subtest (latency)		-	Story content, lexicon and syntax	0.3062 [−0.0251, 0.6375]	0.1691
					Behavioural Inhibition	Tapping		-	Story content, lexicon and syntax	0.1861 [−0.1452, 0.5174]	0.1691
					Behavioural Inhibition	Tapping (latency)		-	Story content, lexicon and syntax	0.2059 [−0.1254, 0.5372]	0.1691
	USA	Typically developing (n = 42)	4.42	4–5	Behavioural Inhibition	Tapping		-	Story content, lexicon and syntax	0.1748 [−0.1391, 0.4886]	0.1600
					Behavioural Inhibition	Tapping (latency)		-	Story content, lexicon and syntax	0.3172 [0.0034, 0.6311]	0.1600
					Shifting	ANT-executive attention subtest		-	Story content, lexicon and syntax	0.3406 [0.0267, 0.6544]	0.1600
					Shifting	ANT-executive attention subtest (latency)		-	Story content, lexicon and syntax	0.009 [−0.3048, 0.3228]	0.1600
Ketelaars et al., 2011 [60]	The Netherlands	Specific Language Impairment (n = 77)	5.60	4–6	-	Nepsy subtests	Oral	Micro-structural	Total Lexical Production	0.3884 [0.1606, 0.6163]	0.1162
	The Netherlands	Typically developing (n = 77)	5.60	4–6	-	Nepsy subtests		Micro-structural	Total Lexical Production	0.3095 [0.0817, 0.5374]	0.1162
Khan, 2013 (dissertation) [51]	USA	Typically developing (n = 84)	4.50	3.5–5	Shifting	Verbal Fluency	Oral	Macro-structural	Story Content	0.2132 [−0.0046, 0.4309]	0.1109
					Planning	Tower of Hanoi			Story Content	0.2769 [0.0591, 0.4946]	0.1109
					Shifting	Card Sorting			Story Content	0.3316 [0.1139, 0.5494]	0.1109
Marini et al., 2020 [96]	Italy	Developmental Language Disorder (n = 16)	5.17	5	Working Memory capacity	Digit Span	Oral	Macro-structural	Information	0.3294 [−0.2142, 0.873]	0.2773
					Interference Control	Square/Circle		Micro-structural	Number Of Utterance	0.5101 [−0.335, 1.0537]	0.2773
						Square/Circle		Macro-structural	Information	0.6169 [0.0734, 1.1605]	0.2773
McNiven, 2010 [55]	Canada	Typically developing (n = 37)	6.95	5–8	Updating of Working Memory	Keep Track	Oral	Macro-structural	Cohesiveness-Referential accuracy	0.3462 [0.0101, 0.6823]	0.1715
					Updating of Working Memory	N-back			Cohesiveness-Referential accuracy	0.362 [0.0259, 0.6982]	0.1715
					Updating of Working Memory	Sound monitoring task			Cohesiveness-Referential accuracy	0.4784 [0.1423, 0.8146]	0.1715
Sacchetti, 2018 (dissertation) [91]	Italy	Typically developing (n = 38–40)	4.92	3–5	Planning	Non-Narrative Sequences	Oral	Micro-structural	Total Lexical Production	0.4392 [0.1079, 0.7705]	0.1691
						Non-Narrative Sequences		Micro-structural	Lexical Variety	0.1186 [−0.2127, 0.4498]	0.1691
						Non-Narrative Sequences		Micro-structural	Morphosyntactic Complexity	0.3417 [0.0104, 0.673]	0.1691
						Non-Narrative Sequences		Micro-structural	Mean Length of Utterance	0.2247 [−0.1066, 0.556]	0.1691
						Non-Narrative Sequences		Macro-structural	Story Content	0.5037 [0.1724, 0.835]	0.1691
						Non-Narrative Sequences		Macro-structural	Coherence of structure	0.5191 [0.1878, 0.8504]	0.1691
					Behavioural Inhibition	Go/NoGo		Micro-structural	Total Lexical Production	0.008 [−0.3142, 0.3302]	0.1643
						Go/NoGo		Micro-structural	Lexical Variety	0.006 [−0.3162, 0.3282]	0.1643
						Go/NoGo		Micro-structural	Morphosyntactic Complexity	0.1034 [−0.2188, 0.4256]	0.1643
						Go/NoGo		Micro-structural	Mean Length of Utterance	0.1409 [−0.1813, 0.4631]	0.1643
						Go/NoGo		Macro-structural	Story Content	0.1419 [−0.1803, 0.4642]	0.1643
						Go/NoGo		Macro-structural	Coherence of structure	0.044 [−0.2782, 0.3662]	0.1643
					Working Memory capacity	Vocal Span		Micro-structural	Total Lexical Production	0.1522 [−0.1701, 0.4744]	0.1643
						Vocal Span		Micro-structural	Lexical Variety	0.1624 [−0.1598, 0.4846]	0.1643
						Vocal Span		Micro-structural	Morphosyntactic Complexity	0.051 [−0.2712, 0.3733]	0.1643
						Vocal Span		Micro-structural	Mean Length of Utterance	0.043 [−0.2792, 0.3652]	0.1643
						Vocal Span		Macro-structural	Information and Story Content	0.0832 [−0.239, 0.4054]	0.1643
						Vocal Span		Macro-structural	Coherence of structure	0.0852 [−0.237, 0.4074]	0.1643
Tonér and Nilsson Gerholm, 2021 [52]	Sweden	Typically developing (n = 47)	5.30	4–6	Interference Control	Flanker	Oral	Micro-structural	Total Lexical Production	0.1409 [−0.1546, 0.4364]	0.1507
					Behavioural Inhibition	Head-Toes-Knees-Shoulders			Total Lexical Production	0.0701 [−0.2254, 0.3656]	0.1507
					Working Memory capacity	Digit Span			Total Lexical Production	0.01 [−0.2855, 0.3055]	0.1507
					Shifting	Dimensional Change Card Sorting			Total Lexical Production	0.01 [−0.2855, 0.3055]	0.1507
					Interference Control	Flanker		Micro-structural	Lexical Variety	0.3654 [0.0700, 0.6609]	0.1507
					Behavioural Inhibition	Head-Toes-Knees-Shoulders			Lexical Variety	0.2132 [−0.0823, 0.5086]	0.1507
					Working Memory capacity	Digit Span			Lexical Variety	0.2554 [−0.041, 0.5509]	0.1507
					Shifting	Dimensional Change Card Sorting			Lexical Variety	0.4847 [0.1892, 0.7802]	0.1507
					Interference Control	Flanker		Micro-structural	Morphosyntactic Accuracy	0.4356 [0.1401, 0.7311]	0.1507
					Behavioural Inhibition	Head-Toes-Knees-Shoulders			Morphosyntactic Accuracy	0.1206 [−0.1749, 0.4161]	0.1507
					Working Memory capacity	Digit Span			Morphosyntactic Accuracy	0.2877 [0.0078, 0.5832]	0.1507
					Shifting	Dimensional Change Card Sorting			Morphosyntactic Accuracy	0.2554 [−0.0401, 0.5509]	0.1507
					Interference Control	Flanker			Morphosyntactic Complexity	0.1614 [−0.1341, 0.4569]	0.1507
					Behavioural Inhibition	Head-Toes-Knees-Shoulders			Morphosyntactic Complexity	0.05 [−0.2454, 0.3455]	0.1507
					Working Memory capacity	Digit Span			Morphosyntactic Complexity	0.2448 [−0.0507, 0.5402]	0.1507
					Shifting	Dimensional Change Card Sorting			Morphosyntactic Complexity	0.3428 [0.0474, 0.6383]	0.1507
					Interference Control	Flanker			Morphosyntactic Complexity–Unified predicates	0.1717 [−0.1238, 0.4671]	0.1507
					Behavioural Inhibition	Head-Toes-Knees-Shoulders			Morphosyntactic Complexity–Unified predicates	0.03 [−0.2655, 0.3255]	0.1507
					Working Memory capacity	Digit Span			Morphosyntactic Complexity–Unified predicates	0.1206 [−0.1749, 0.4161]	0.1507
					Shifting	Dimensional Change Card Sorting			Morphosyntactic Complexity–Unified predicates	0.3316 [0.0362, 0.6271]	0.1507
					Interference Control	Flanker		Macro-structural	Information	0.2877 [−0.0078, 0.5832]	0.1507
					Behavioural Inhibition	Head-Toes-Knees-Shoulders			Information	0.1104 [−0.185, 0.4059]	0.1507
					Working Memory capacity	Digit Span			Information	0.3095 [0.0140, 0.6050]	0.1507
					Shifting	Dimensional Change Card Sorting			Information	0.4722 [0.1768, 0.7677]	0.1507
Veraksa et al., 2020 [93]	Russia	Typically developing (n = 269)	5.58	5–6	Working Memory capacity	Memory Design	Oral	Micro-structural	Morphosyntactic Accuracy	0.1206 [0.0004, 0.2408]	0.0616
						Memory Design		Micro-structural	Number Of Syntagmas	0.1511 [0.0310, 0.2713]	0.0616
						Memory Design		Micro-structural	Number Of Simple Utterance	0.1511 [0.0310, 0.2713]	0.0616
						Memory Design		Macro-structural	Coherence–Semantic adequacy	0.1614 [0.0412, 0.2816]	0.0616
						Memory Design		Micro-structural	Lexical Production	0.1614 [0.412, 0.2816]	0.0616
						Memory Design		Macro-structural	Coherence–programming	0.182 [0.0618, 0.3022]	0.0616
					Working Memory capacity	Sentence Repetition		Micro-structural	Number Of Simple Utterance	0.2027 [0.0826, 0.3229]	0.0616
						Sentence Repetition			Number Of Syntagmas	0.2237 [0.1035, 0.3438]	0.0616
					Working Memory capacity	Memory Design		Macro-structural	Coherence–Semantic completeness	0.2342 [0.114, 0.3544]	0.0616
						Memory Design			Coherence of structure	0.2554 [0.1352, 0.3756]	0.0616
					Working Memory capacity	Sentence Repetition		Micro-structural	Total Lexical Production	0.2554 [0.1352, 0.3756]	0.0616
					Working Memory capacity	Memory Design		Macro-structural	Coherence–narrative structure	0.2661 [0.1459, 0.3863]	0.0616
					Working Memory capacity	Sentence Repetition		Micro-structural	Morphosyntactic Accuracy	0.3205 [0.2004, 0.4407]	0.0616
						Sentence Repetition		Macro-structural	Coherence–Semantic adequacy	0.4356 [0.3154, 0.5558]	0.0616
						Sentence Repetition			Coherence–narrative structure	0.4599 [0.3397, 0.5801]	0.0616
						Sentence Repetition			Coherence–programming	0.4847 [0.3645, 0.6049]	0.0616
						Sentence Repetition			Coherence–narrative type (complete, simplified, distorted)	0.5361 [0.4159, 0.6562]	0.0616
						Sentence Repetition			Coherence–Semantic completeness	0.5493 [0.4291, 0.6695]	0.0616

**Table 4 children-10-01391-t004:** Studies including participants aged 8–18 year old.

References	Location	Clinical Risk Status of the Sample	Mean Age (Years)	Age Range	EF Domain	EF Task	Narrative Form	Narrative Competence	NC Indicator	Fisher’s Z [95% CI]	SE
Artico and Penge, 2016 [64]	Italy	Dyslexia and Dysgraphia (n = 54)	9.87	8–12	Shifting	Verbal Fluency	Written	Micro-structural	Lexical Variety	0.1003 [0.1741, 0.3748]	0.1400
						Verbal Fluency		Macro-structural	Cohesiveness	0.1003 [−0.1741, 0.3748]	0.1400
					Planning	Tower of London		Micro-structural	Morphosyntactic Complexity	0.1307 [−0.1437, 0.4052]	0.1400
					Shifting	Response set (NEPSY II)		Micro-structural	Total Lexical Production	0.1409 [−0.1335, 0.4154]	0.1400
					Planning	Tower of London			Total Lexical Production	0.1409 [−0.1335, 0.4154]	0.1400
					Shifting	Verbal Fluency			Total Lexical Production	0.1717 [−0.1028, 0.4461]	0.1400
					Planning	Tower of London		Micro-structural	Lexical Variety	0.1820 [−0.0925, 0.4564]	0.1400
					Shifting	Response set (NEPSY II)		Macro-structural	Coherence	0.1820 [−0.0925, 0.4564]	0.1400
					Shifting	Switching NEPSY II		Macro-structural	Cohesiveness	0.1923 [−0.0821, 0.4668]	0.1400
					Planning	Tower of London			Cohesiveness	0.1923 [−0.0821, 0.4668]	0.1400
					Shifting	Switching NEPSY II		Micro-structural	Total Lexical Production	0.2027 [−0.0717, 0.4772]	0.1400
					Shifting	Response set (NEPSY II)		Micro-structural	LexicalVariety	0.2132 [−0.0613, 0.4876]	0.1400
					Shifting	Verbal Fluency		Macro-structural	Coherence	0.2132 [−0.0613, 0.4876]	0.1400
					Planning	Tower of London			Coherence	0.2132 [−0.0613, 0.4876]	0.1400
					Planning	Clocks		Macro-structural	Cohesiveness	0.2342 [0.4030, 0.5086]	0.1400
					Shifting	Switching NEPSY II		Macro-structural	Coherence	0.2342 [0.4030, 0.5086]	0.1400
					Shifting	Response set (NEPSY II)		Macro-structural	Cohesiveness	0.2448 [−0.0297, 0.5192]	0.1400
					Behavioural Inhibition	Go/NoGo		Macro-structural	Coherence	0.2448 [−0.0297, 0.5192]	0.1400
					Planning	Clocks		Micro-structural	Total Lexical Production	0.2877 [0.0132, 0.5621]	0.1400
					Shifting	Switching NEPSY II		Micro-structural	Lexical Variety	0.2986 [0.0241, 0.5730]	0.1400
					Shifting	Response set (NEPSY II)		Micro-structural	Morphosyntactic Complexity	0.2986 [0.0241, 0.5730]	0.1400
					Behavioural Inhibition	Go/NoGo		Micro-structural	Total Lexical Production	0.3428 [0.0684, 0.6173]	0.1400
					Shifting	Verbal Fluency		Micro-structural	Morphosyntactic Complexity	0.3541 [0.0796, 0.6285]	0.1400
					Behavioural Inhibition	Go/NoGo		Macro-structural	Cohesiveness	0.3541 [0.0796, 0.6285]	0.1400
						Go/NoGo		Micro-structural	Lexical Variety	0.3654 [0.0910, 0.6399]	0.1400
					Planning	Clocks		Micro-structural	Morphosyntactic Complexity	0.3884 [0.1140, 0.6629]	0.1400
						Clocks		Macro-structural	Coherence	0.4001 [0.1256, 0.6745]	0.1400
						Clocks		Micro-structural	Lexical Variety	0.4236 [0.1492, 0.6981]	0.1400
					Shifting	Switching NEPSY II		Micro-structural	Morphosyntactic Complexity	0.4599 [0.1854, 0.7343]	0.1400
					Behavioural Inhibition	Go/NoGo			Morphosyntactic Complexity	0.5230 [0.2485, 0.7974]	0.1400
Balaban et al., 2020 [53]	Turkia	Typically Developing (n = 87)	8.17	7–11	Behavioural Inhibition	Emotional Stroop Task	Oral	Micro-structural	Syntactic Complexity	0.1717 [−0.0422, 0.3855]	0.1091
						Emotional Stroop Task		Macro-structural	Plot Complexity	0.3316 [0.1178, 0.5455]	0.1091
Balioussis et al., 2012 [56]	Canada	Typically Developing (n = 70)	9.83	8–9	Working Memory capacity	Letter Memory Task	Written	Micro-structural	Morphosyntactic Complexity	0.3541 [0.1146, 0.5935]	0.1221
					Shifting	Contingency Naming Task		Micro-structural	Total Lexical Production	0.4599 [0.2204, 0.6993]	0.1221
					Working Memory capacity	Letter Memory Task			Total Lexical Production	0.3316 [0.0922, 0.5711]	0.1221
					Shifting	Contingency Naming Task		Micro-structural	Syntactic Complexity	0.3428 [0.1034, 0.5823]	0.1221
Drijbooms et al., 2017 [54]	The Netherlands	Typically Developing (n = 93)	11.08	-	-	Trail Making Test; Tower of London	Written	Micro-structural	Total Lexical Production	0.03 [−0.1766, 0.2366]	0.1054
					-	Trail Making Test; Tower of London		Macro-structural	Story content	0.03 [−0.1766, 0.2366]	0.1054
					-	Digit Span; Letter Fluency; Ricerca visiva			Story content	0.0601 [−0.1465, 0.2667]	0.1054
					-	Digit Span; Letter Fluency; Ricerca visiva		Micro-structural	Morphosyntactic Complexity	0.0701 [−1365, 0.2767]	0.1054
					-	Digit Span; Letter Fluency; Ricerca visiva		Micro-structural	Total Lexical Production	0.0701 [−0.1365, 0.2767]	0.1054
					-	Walk Don’t Walk; Opposite Worlds; Trail Making Test; Letter Digit Substitution			Total Lexical Production	0.1717 [−0.0349, 0.3783]	0.1054
					-	Walk Don’t Walk; Opposite Worlds; Trail Making Test; Letter Digit Substitution		Macro-structural	Story content	0.2027 [−0.0039, 0.4093]	0.1054
					-	Trail Making Test; Tower of London		Micro-structural	Morphosintactic Complexity	0.2237 [0.0171, 0.4303]	0.1054
					-	Walk Don’t Walk; Opposite Worlds; Trail Making Test; Letter Digit Substitution			Morphosintactic Complexity	0.2554 [0.0488, 0.462]	0.1054
Drijbooms et al., 2015 [61]	The Netherlands	Typically Developing (n = 102)	9.58	8–11	Planning	Tower of London	Written	Micro-structural	Morphosyntactic Complexity	0.05 [−0.1469, 0.247]	0.1005
					Shifting	Trail Making Test		Micro-structural	Total Lexical Production	0.0701 [−0.1269, 0.2671]	0.1005
					Planning	Tower of London			Total Lexical Production	0.0701 [−0.1269, 0.2671]	0.1005
					Behavioural Inhibition	Opposite words		Macro-structural	Story content	0.1003 [−0.0966, 0.2973]	0.1005
					Shifting	Trail Making Test		Micro-structural	Morphosyntactic Complexity	0.1104 [−0.0865, 0.3074]	0.1005
					Working Memory capacity	Digit Span		Macro-structural	StoryContent	0.1409 [−0.0561, 0.3379]	0.1005
						Digit Span		Micro-structural	Total Lexical Production	0.1511 [−0.0458, 0.3481]	0.1005
					Planning	Tower of London		Macro-structural	Story content	0.1511 [−0.0458, 0.3481]	0.1005
					Behavioural Inhibition	Walk don’t Walk			Story content	0.1717 [−0.0253, 0.3687]	0.1005
					Shifting	Trail Making Test			Story content	0.1717 [−0.0253, 0.3687]	0.1005
					Behavioural Inhibition	Walk don’t Walk		Micro-structural	Morphosintactic Complexity	0.182 [−0.015, 0.379]	0.1005
					Behavioural Inhibition	Opposite words			Morphosyntactic Complexity	0.2132 [0.0162, 0.4102]	0.1005
					Working Memory capacity	Digit Span			Morphosyntactic Complexity	0.2237 [0.0267, 0.4206]	0.1005
					Behavioural Inhibition	Opposite words		Micro-structural	Total Lexical Production	0.2448 [0.0478, 0.4418]	0.1005
					Behavioural Inhibition	Walk don’t Walk			Total Lexical Production	0.2554 [0.0584, 0.4524]	0.1005
Fisher et al., 2019 [97]	USA	Dyslexia (n = 92)	9.25	-	Shifting	Card Sorting	Oral	Macro-structural	Coherence	0.1206 [−0.0872, 0.3283]	0.1058
					Interference Control	Stroop			Coherence	0.1614 [−0.0464, 0.3691]	0.1058
					Shifting	Trail Making Test			Coherence	0.1923 [−0.0154, 0.4001]	0.1058
					Working Memory capacity	Corsi			Coherence	0.2877 [0.0799, 0.4954]	0.1058
Park, 2014 (dissertation) [80]	USA	Typically Developing (n = 10)	10.00	9–11	Shifting	Trail Making Test	Oral	Macro-structural	GAO units	0.4611 [−0.2797, 1.2019]	0.3780
						Trail Making Test		Macro-structural	Complete GAO units (Integrity)	0.1318 [−0.609, 0.8726]	0.3780
					Planning	Tower of London			Complete GAO units (Integrity)	0.0993 [−0.6415, 0.8401]	0.3780
						Tower of London		Macro-structural	GAO units–episodic structure	0.038 [−0.7028, 0.7788]	0.3780
					Shifting	Card Sorting		Macro-structural	Complete GAO units (Integrity)	0.2079 [−0.5329, 0.9487]	0.3780
					Working Memory capacity	Digit Span Backword			Complete GAO units (Integrity)	0.2586 [−0.4822, 0.9994]	0.3780
						Digit Span Backword		Macro-structural	GAO units	0.5682 [−0.1726, 1.3089]	0.3780
					Shifting	Card Sorting			GAO units	0.8053 [0.0645, 1.5461]	0.3780
		Deaf or hard to hearing (n = 11)	10.00	9–11	Planning	Tower of London	Oral	Macro-structural	GAO units	0.5874 [−0.1056, 1.2803]	0.3536
					Working Memory capacity	Digit Span Backword			GAO units	0.3451 [−0.3479, 1.038]	0.3536
					Shifting	Card Sorting			GAO units	0.2384 [−0.4545, 0.9314]	0.3536
					Working Memory capacity	Digit Span Backword		Macro-structural	Complete GAO units (Integrity)	0.1145 [−0.5785, 0.8074]	0.3536
					Planning	Tower of London			Complete GAO units (Integrity)	0.1155 [−0.5774, 0.8085]	0.3536
					Shifting	Trail Making Test			Complete GAO units (Integrity)	0.1348 [−0.5581, 0.8278]	0.3536
						Trail Making Test		Macro-structural	GAO units	0.231 [−0.4619, 0.924]	0.3536
					Shifting	Card Sorting		Macro-structural	Complete GAO units (Integrity)	0.4047 [−0.2882, 1.0977]	0.3536
Peristeri et al., 2020 [79]	Greece	Autism Spectrum Disorder (n = 20)	9.80	7–12	Updating of Working Memory	2-back	Oral	Micro-structural	Lexical Variety	0.1246 [−0.3507, 0.6]	0.2425
						2-back		Micro-structural	Morphosyntactic Complexity	0.1522 [−0.3232, 0.6275]	0.2425
						2-back		Micro-structural	Number of subordinated clauses	0.2501 [−0.2253, 0.7254]	0.2425
						2-back		Micro-structural	Number of relative clauses	0.046 [−0.4293, 0.5214]	0.2425
						2-back		Macro-structural	Story Structure	0.146 [−0.3293, 0.6214]	0.2425
						2-back		Macro-structural	Referential Accuracy	0.4153 [−0.06, 0.8907]	0.2425
					Interference Control	Local-to-Global (Accuracy)		Micro-structural	Lexical Variety	0.0993 [−0.376, 0.5747]	0.2425
						Local-to-Global (Accuracy)		Micro-structural	Morphosyntactic Complexity	0.3272 [0.1482, 0.8026]	0.2425
						Local-to-Global (Accuracy)		Micro-structural	Number of subordinated clauses	0.031 [−0.4444, 0.5064]	0.2425
						Local-to-Global (Accuracy)		Micro-structural	Number of relative clauses	0.047 [−0.4283, 0.5224]	0.2425
						Local-to-Global (Accuracy)		Macro-structural	Story Structure	0.0591 [−0.4163, 0.5344]	0.2425
						Local-to-Global (Accuracy)		Macro-structural	Referential Accuracy	0.353 [−0.1224, 0.8283]	0.2425
					Interference Control	Global-to-Local (Accuracy)		Micro-structural	Lexical Variety	0.1206 [−0.3548, 0.5959]	0.2425
						Global-to-Local (Accuracy)		Micro-structural	Morphosyntactic Complexity	0.0621 [−0.4133, 0.5374]	0.2425
						Global-to-Local (Accuracy)		Micro-structural	Number of subordinated	0.0902 [−0.3851, 0.5656]	0.2425
						Global-to-Local (Accuracy)		Micro-structural	Number of relatives	0.019 [−0.4564, 0.4944]	0.2425
						Global-to-Local (Accuracy)		Macro-structural	Story Structure	0.4822 [0.0068, 0.9576]	0.2425
						Global-to-Local (Accuracy)		Macro-structural	Referential Accuracy	0.0661 [−0.4093, 0.5415]	0.2425
					Interference Control	Local-to-Global (Reaction Time)		Micro-structural	Lexical Variety	0.4562 [−0.0191, 0.9316]	0.2425
						Local-to-Global (Reaction Time)		Micro-structural	Morphosyntactic Complexity	0.3598 [−0.1156, 0.8351]	0.2425
						Local-to-Global (Reaction Time)		Micro-structural	Number of subordinated clauses	0.2942 [−0.1812, 0.7696]	0.2425
						Local-to-Global (Reaction Time)		Micro-structural	Number of relative clauses	0.3372 [−0.1381, 0.8126]	0.2425
						Local-to-Global (Reaction Time)		Macro-structural	Story Structure	0.037 [−0.4383, 0.5124]	0.2425
						Local-to-Global (Reaction Time)		Macro-structural	Referential Accuracy	0.049 [−0.4263, 0.5244]	0.2425
					Interference Control	Global-to-Local (Reaction Time)		Micro-structural	Lexical Variety	0.4648 [−0.0105, 0.9402]	0.2425
						Global-to-Local (Reaction Time)		Micro-structural	Morphosyntactic Complexity	0.2715 [−0.2039, 0.7468]	0.2425
						Global-to-Local (Reaction Time)		Micro-structural	Number of subordinated clauses	0.1013 [−0.374, 0.5767]	0.2425
						Global-to-Local (Reaction Time)		Micro-structural	Number of relative clauses	0.045 [−0.4303, 0.5204]	0.2425
						Global-to-Local (Reaction Time)		Macro-structural	Story Structure	0.482 [0.0068, 0.9576]	0.2425
						Global-to-Local (Reaction Time)		Macro-structural	Referential Accuracy	0.0661 [−0.4093, 0.5415]	0.2425
Peristeri et al., 2020 [79]	Greece	Typically Developing (n = 20)	9.80	7–12	Updating of Working Memory	2-back	Oral	Micro-structural	Lexical Variety	0.1257 [−0.3497, 0.601]	0.2425
						2-back		Micro-structural	Morphosyntactic Complexity	0.0862 [−0.3891, 0.5616]	0.2425
						2-back		Micro-structural	Number of subordinated clauses	0.2048 [−0.2705, 0.6802]	0.2425
						2-back		Micro-structural	Number of relative clauses	0.146 [−0.3293, 0.6214]	0.2425
						2-back		Macro-structural	Story Structure	0.1064 [−0.369, 0.5818]	0.2425
						2-back		Macro-structural	Referential Accuracy	0.231 [−0.2443, 0.7064]	0.2425
					Interference Control	Local-to-Global (Accuracy)		Micro-structural	Lexical Variety	0.0621 [−0.4133, 0.5374]	0.2425
						Local-to-Global (Accuracy)		Micro-structural	Morphosyntactic Complexity	0.2779 [−0.1974, 0.7533]	0.2425
						Local-to-Global (Accuracy)		Micro-structural	Number of subordinated clauses	0.045 [−0.4303, 0.5204]	0.2425
						Local-to-Global (Accuracy)		Micro-structural	Number of relative clauses	0.9417 [0.4663, 1.4171]	0.2425
						Local-to-Global (Accuracy)		Macro-structural	Story Structure	0.2342 [−0.2412, 0.7096]	0.2425
						Local-to-Global (Accuracy)		Macro-structural	Referential Accuracy	0.1389 [−0.3365, 0.6142]	0.2425
					Interference Control	Global-to-Local (Accuracy)		Micro-structural	Lexical Variety	0.041 [−0.4343, 0.5164]	0.2425
						Global-to-Local (Accuracy)		Micro-structural	Morphosyntactic Complexity	0.5139 [0.0386, 0.9893]	0.2425
						Global-to-Local (Accuracy)		Micro-structural	Number of subordinated clauses	0.0923 [−0.3831, 0.5676]	0.2425
						Global-to-Local (Accuracy)		Micro-structural	Number of relative clauses	0.7137 [0.2384, 1.1891]	0.2425
						Global-to-Local (Accuracy)		Macro-structural	Story Structure	0.3496 [−0.1258, 0.8249]	0.2425
						Global-to-Local (Accuracy)		Macro-structural	Referential Accuracy	0.0701 [−0.4052, 0.5455]	0.2425
					Interference Control	Local-to-Global (Reaction Time)		Micro-structural	Lexical Variety	1.211 [0.7357, 1.6864]	0.2425
						Local-to-Global (Reaction Time)		Micro-structural	Morphosyntactic Complexity	0.5308 [0.0554, 1.0062]	0.2425
						Local-to-Global (Reaction Time)		Micro-structural	Number of subordinated clauses	0.2877 [−0.1877, 0.763]	0.2425
						Local-to-Global (Reaction Time)		Micro-structural	Number of relative clauses	0.3507 [−0.1247, 0.8261]	0.2425
						Local-to-Global (Reaction Time)		Macro-structural	Story Structure	0.7582 [0.2828, 1.2335]	0.2425
						Local-to-Global (Reaction Time)		Macro-structural	Referential Accuracy	0.2533 [−0.2221, 0.7286]	0.2425
					Interference Control	Global-to-Local (Reaction Time)		Micro-structural	Lexical Variety	0.1206 [0.3548, 0.5959]	0.2425
						Global-to-Local (Reaction Time)		Micro-structural	Morphosyntactic Complexity	0.0741 [−0.4012, 0.5495]	0.2425
						Global-to-Local (Reaction Time)		Micro-structural	Number of subordinated clauses	0.1186 [−0.3568, 0.5939]	0.2425
						Global-to-Local (Reaction Time)		Micro-structural	Number of relative clauses	0.3586 [−0.1167, 0.834]	0.2425
						Global-to-Local (Reaction Time)		Macro-structural	Story Structure	0.6155 [0.1402, 1.0909]	0.2425
						Global-to-Local (Reaction Time)		Macro-structural	Referential Accuracy	0.002 [−0.4734, 0.4774]	0.2425
Puranik, 2006 (dissertation) [57]	USA	Typically Developing (n = 90)	10.22	8–12	Working Memory capacity	Competing Language Processing Task	Written	Micro-structural	Total Lexical Production	0.4001 [0.1899, 0.6102]	0.1072
					Working Memory capacity	Digit Ordering			Total Lexical Production	0.3316 [0.1215, 0.5418]	0.1072
					Working Memory capacity	Competing Language Processing Task		Macro-structural	Information	0.4118 [0.2017, 0.6219]	0.1072
					Working Memory capacity	Digit Ordering			Information	0.3884 [0.1783, 0.5986]	0.1072
					Working Memory capacity	Competing Language Processing Task		Micro-structural	Number of Utterance	0.2986 [0.0884, 0.5087]	0.1072
					Working Memory capacity	Digit Ordering			Number of Utterance	0.2661 [0.056, 0.4762]	0.1072
Salas and Silvente, 2020	Spain	Typically Developing (n = 1337)	10.17	7–14	Interference Control	Stroop	Written	Micro-structural	Mean Length of Utterance	0.0802 [0.0265, 0.1338]	0.0265
					Working Memory capacity	Digit Span		Micro-structural	Total Lexical Production	0.2237 [0.17, 0.2773]	0.0265
					Working Memory capacity	Digit Span		Micro-structural	Mean Length of Utterance	0.0802 [0.0265, 0.1338]	0.0265
					Interference Control	Stroop		Micro-structural	Total Lexical Production	0.2342 [0.1805, 0.2879]	0.0265
Swanson and Berninger, 1996a [63]	USA	Typically Developing (n = 300)	11.09	9–12	Working Memory capacity	Listening Recall, Listening Generate Recall	Written	Micro-structural	Number of Utterance	0.2769 [0.1631, 0.3906]	0.0583
						Listening Recall, Listening Generate Recall		Macro-structural	Content and organization	0.2554 [0.1417, 0.3691]	0.0583
					Working Memory capacity	Matrix		Micro-structural	Number of Utterance	0.0601 [−0.0537, 0.1738]	0.0583
						Matrix		Macro-structural	Content and organization	0.1206 [0.0069, 0.2343]	0.0583
Swanson and Berninger, 1996b [58]	USA	Typically Developing (n = 50)	10.50	9–12	Working Memory capacity	Sentence Span Test	Written	Macro-structural	Content	0.3095 [0.0236, 0.5945]	0.1459
						Sentence Span Test		Micro-structural	Mean Length of Utterance	0.2769 [−0.009, 0.5628]	0.1459
						Sentence Span Test		Micro-structural	Total Lexical Production	0.3654 [0.0796, 0.6513]	0.1459
Vanderberg and Swanson, 2006 [94]	USA	Typically Developing (n = 160)	15.21	14–15	Working Memory capacity	Rhyming words	Written	Macro-structural	Structure	0.182 [0.0256, 0.3384]	0.0800
						Rhyming words		Micro-structural	Total Lexical Production	0.1511 [−0.0053, 0.3076]	0.0800
						Rhyming words		Micro-structural	Morphosyntactic Complexity	0.0902 [−0.0662, 0.2467]	0.0800
					Working Memory capacity	Sentence Span		Macro-structural	Structure	0.1104 [−0.046, 0.2669]	0.0800
						Sentence Span		Micro-structural	Total Lexical Production	0.0701 [−0.0863, 0.2265]	0.0800
						Sentence Span		Micro-structural	Morphosyntactic Complexity	0.1409 [−0.0155, 0.2973]	0.0800
					Working Memory capacity	Visual Matrix		Macro-structural	Structure	0.0902 [−0.0662, 0.2467]	0.0800
						Visual Matrix		Micro-structural	Total Lexical Production	0.1409 [−0.0155, 0.2973]	0.0800
						Visual Matrix		Micro-structural	Morphosyntactic Complexity	0.0601 [−0.0964, 0.2165]	0.0800
					Working Memory capacity	Mapping		Macro-structural	Structure	0.01 [−0.1464, 0.1664]	0.0800
						Mapping		Micro-structural	Total Lexical Production	−0.0601 [−0.2165, 0.0964]	0.0800
						Mapping		Micro-structural	Morphosyntactic Complexity	0.02 [−0.1364, 0.1764]	0.0800
Fernandez et al., 2010 [66]	Spain	Attention Deficit Hyperactivity Disorder (n = 26)	8.50	6–11	Behavioural Inhibition	Matching Familiar Figure Test	Oral	Macro-structural	Coherence	0.4236 [0.015, 0.8323]	0.2086
					Working Memory capacity	Digit Span	Oral		Coherence	0.1104 [−0.2982, 05191]	0.2086
					Interference Control	Stroop	Oral		Coherence	0.2661 [−0.1426, 0.6748]	0.2086
					Working Memory capacity	Rey Figure	Oral		Coherence	0.4973 [0.0886, 0.906]	0.2086

## Data Availability

Not applicable.

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
