# Peer review of "The Relationship between Narrative Skills and Executive Functions across Childhood: A Systematic Review and Meta-Analysis"

_children, 2023, doi:10.3390/children10081391_

Round 1
Reviewer 1 Report
The authors should be commended for undertaking a rigorous meta-analysis on the relationship between narrative skills (NS) and executive functions (EF). The rationale for the study was clearly stated namely exploring the relationship between narrative skills and executive functioning across childhood and adolescence. Analysis of relevant studies was preceded by an extensive critical review of the literature highlighting the strength and limitations in the context of the theme. And as they pointed out 'there is conflicting evidence about the developmental stages at which EF relates to NC'. Their findings are clearly outlined followed by a thorough discussion of their implications. The authors have achieved their objective by acknowledging the limits of their findings followed by elaborating areas that require further investigation. They have not hesitated to propose hypotheses for some of the areas which require further investigation. Recommendations of areas for future research at the end of the article clearly provide a concise future agenda emerging from their study. The authors should be congratulated for a remarkable piece of work. There are two typos sentence 764 the word 'Togheter' is meant to be 'Together'. Sentence 862 'It would be insightful [to] observe...', please insert 'to'.
Author Response
We appreciated very much the comments and we corrected the sentences cited by the reviewer.
Point 1. There are two typos sentence 764 the word 'Togheter' is meant to be 'Together'. Sentence 862 'It would be insightful [to] observe...', please insert 'to'.
Response 1. We corrected the two typos.
Reviewer 2 Report
Multiple domains would have been considered.
Authors need to justify how two different cognitive domains can be dissociated? Physiological explanation is required.
Overall quality of presentation good.
Authors have done satisfactory work
Author Response
Point 1: Authors need to justify how two different cognitive domains can be dissociated? Physiological explanation is required.
Response 1: In the introduction we specified the neurological distinction between narrative competences and executive function.
"However, it is possible to find specific brain regions associated with narrative competence such as temporal poles, the posterior cingulate, and the left superior temporal gyrus (Fletcher et al, 1995). On the other hand, cognitive executive functions are more associated with bilateral dorsolateral prefrontal cortex (Diamond, 2013)."
Reviewer 3 Report
I found a very interesting work in the manuscript: The Relation Between Narrative Skills and Executive Functions Across Childhood: A Systematic Review and Meta-Analysis.
This meta-analysis presents a relevant topic in children's development and neuroscience. Authors made a scholar introduction and description of their variables of interest.
Two minor comments as suggestions for improving their work:
Introduction, the aims of the study are clearly listed at the end of this section, however, there are no previous research questions stated or introduced along the mentioned section. For example, the authors refer to autism but there are no issues related to this neurological condition during the introduction.
Method. I suggest authors to state in the methods section about th novelty of their search strategy, and if there are no related strategies registered in PROSPERO. Apendix A (strings) mention as exclusion criteria NOT review, but Exclusion criteria section do not mention if authors excluded systematic reviews and meta-analysis from the sample.
Was there any consideration regarding the native language from the sample as part of possible bias?
Author Response
Point 1: Introduction, the aims of the study are clearly listed at the end of this section, however, there are no previous research questions stated or introduced along the mentioned section. For example, the authors refer to autism but there are no issues related to this neurological condition during the introduction.
Response 1: At the end of the Introduction we cited some neurodevelopmental disorders (e.g. Autism) but we have not specified each disorder, firstly because it's not the focus of this paper, secondly because there is not enough space, finally because we compared typical vs atypical children without distinguishing each neurodevelopmental disorder.
Point 2: Method. I suggest authors to state in the methods section about the novelty of their search strategy, and if there are no related strategies registered in PROSPERO. Apendix A (strings) mention as exclusion criteria NOT review, but Exclusion criteria section do not mention if authors excluded systematic reviews and meta-analysis from the sample.
Response 2: We checked in Prospero and we confirmed that there are no other study similar to this one. For this reason we added this sentence in the Method section "This is the first meta-analysis on Narrative Competence and Executive Function in children and adolescents."
Moreover, we have inserted the instruction "NOT Review" in order to be formally correct but there are not published reviews on Narrative Competence and Executive Function.